# Online Learning in Episodic Markovian Decision Processes by Relative Entropy Policy Search

**Alexander Zimin**
Institute of Science and Technology Austria
alexander.zimin@ist.ac.at

**Gergely Neu**
INRIA Lille – Nord Europe
gergely.neu@gmail.com

## Abstract

We study the problem of online learning in finite episodic Markov decision processes (MDPs) where the loss function is allowed to change between episodes. The natural performance measure in this learning problem is the regret defined as the difference between the total loss of the best stationary policy and the total loss suffered by the learner. We assume that the learner is given access to a finite action space $\mathcal{A}$ and the state space $\mathcal{X}$ has a layered structure with $L$ layers, so that state transitions are only possible between consecutive layers. We describe a variant of the recently proposed Relative Entropy Policy Search algorithm and show that its regret after $T$ episodes is $2\sqrt{L|\mathcal{X}||\mathcal{A}|T \log(|\mathcal{X}||\mathcal{A}|/L)}$ in the bandit setting and $2L\sqrt{T \log(|\mathcal{X}||\mathcal{A}|/L)}$ in the full information setting, given that the learner has perfect knowledge of the transition probabilities of the underlying MDP. These guarantees largely improve previously known results under much milder assumptions and cannot be significantly improved under general assumptions.

## 1 Introduction

In this paper, we study the problem of online learning in a class of finite non-stationary episodic Markov decision processes. The learning problem that we consider can be formalized as a sequential interaction between a *learner* (often called *agent*) and an *environment*, where the interaction between the two entities proceeds in *episodes*. Every episode consists of multiple time steps: In every time step of an episode, a learner has to choose one of its available actions after observing some part of the current state of the environment. The chosen action influences the observable state of the environment in a stochastic fashion and imposes some loss on the learner. However, the entire state (be it observed or not) also influences the loss. The goal of the learner is to minimize its total (non-discounted) loss that it suffers. In this work, we assume that the unobserved part of the state evolves autonomously from the observed part of the state or the actions chosen by the learner, thus corresponding to a state sequence generated by an oblivious adversary such as nature. Otherwise, absolutely no statistical assumption is made about the mechanism generating the unobserved state variables. As usual for such learning problems, we set our goal as minimizing the regret defined as the difference between the total loss suffered by the learner and the total loss of the best stationary state-feedback policy. This setting fuses two important paradigms of learning theory: online learning [5] and reinforcement learning [21, 22].

The learning problem outlined above can be formalized as an online learning problem where the actions of the learner correspond to choosing policies in a known Markovian decision process where the loss function changes arbitrarily between episodes. This setting is a simplified version of the

Parts of this work were done while Alexander Zimin was enrolled in the MSc. programme of the Central European University, Budapest, and Gergely Neu was working on his PhD. thesis at the Budapest University of Technology and Economics and the MTA SZTAKI Institute for Computer Science and Control, Hungary. Both authors would like to express their gratitude to László Györfi for making this collaboration possible.

learning problem first addressed by Even-Dar et al. [8, 9], who consider online learning unichain MDPs. In their variant of the problem, the learner faces a continuing MDP task where all policies are assumed to generate a unique stationary distribution over the state space and losses can change arbitrarily between consecutive time steps. Assuming that the learner observes the complete loss function after each time step (that is, assuming *full information feedback*), they propose an algorithm called MDP-E and show that its regret is $O(\tau^2\sqrt{T\log|\mathcal{A}|})$, where $\tau > 0$ is an upper bound on the mixing time of any policy. The core idea of MDP-E is the observation that the regret of the global decision problem can be decomposed into regrets of simpler decision problems defined in each state. Yu et al. [23] consider the same setting and propose an algorithm that guarantees $o(T)$ regret under *bandit feedback* where the learner only observes the losses that it actually suffers, but not the whole loss function. Based on the results of Even-Dar et al. [9], Neu et al. [16] propose an algorithm that is shown to enjoy an $O(T^{2/3})$ bound on the regret in the bandit setting, given some further assumptions concerning the transition structure of the underlying MDP. For the case of continuing *deterministic* MDP tasks, Dekel and Hazan [7] describe an algorithm guaranteeing $O(T^{2/3})$ regret. The immediate precursor of the current paper is the work of Neu et al. [14], who consider online learning in episodic MDPs where the state space has a layered (or *loop-free*) structure and *every policy visits every state with a positive probability of at least* $\alpha > 0$. Their analysis is based on a decomposition similar to the one proposed by Even-Dar et al. [9], and is sufficient to prove a regret bound of $O(L^2\sqrt{T|\mathcal{A}|\log|\mathcal{A}|}/\alpha)$ in the bandit case and $O(L^2\sqrt{T\log|\mathcal{A}|})$ in the full information case.

In this paper, we present a learning algorithm that directly aims to minimize the global regret of the algorithm instead of trying to minimize the local regrets in a decomposed problem. Our approach is motivated by the insightful paper of Peters et al. [17], who propose an algorithm called *Relative Entropy Policy Search* (REPS) for reinforcement learning problems. As Peters et al. [17] and Kakade [11] point out, good performance of policy search algorithms requires that the *information loss* between the consecutive policies selected by the algorithm is bounded, so that policies are only modified in small steps. Accordingly, REPS aims to select policies that minimize the expected loss while guaranteeing that the state-action distributions generated by the policies stay close in terms of Kullback–Leibler divergence. Further, Daniel et al. [6] point out that REPS is closely related to a number of previously known probabilistic policy search methods. Our paper is based on the observation that REPS is closely related to the Proximal Point Algorithm (PPA) first proposed by Martinet [13] (see also [20]).

We propose a variant of REPS called online REPS or O-REPS and analyze it using fundamental results concerning the PPA family. Our analysis improves all previous results concerning online learning in episodic MDPs: we show that the expected regret of O-REPS is bounded by $2\sqrt{L|\mathcal{X}||\mathcal{A}|T\log(|\mathcal{X}||\mathcal{A}|/L)}$ in the bandit setting and $2L\sqrt{T\log(|\mathcal{X}||\mathcal{A}|/L)}$ in the full information setting. Unlike previous works in the literature, we do not have to make any assumptions about the transition dynamics apart from the loop-free assumption. The full discussion of our results is deferred to Section 5.

Before we move to the technical content of the paper, we first fix some conventions. Random variables will be typeset in boldface (e.g., $\mathbf{x}$, $\mathbf{a}$) and indefinite sums over states and actions are to be understood as sums over the entire state and action spaces. For clarity, we assume that all actions are available in all states, however, this assumption is not essential. The indicator of any event $A$ will be denoted by $\mathbb{I}\{A\}$.

## 2 Problem definition

An *episodic loop-free Markov decision process* is formally defined by the tuple $M = \{\mathcal{X}, \mathcal{A}, P\}$, where $\mathcal{X}$ is the finite state space, $\mathcal{A}$ is the finite action space, and $P : \mathcal{X} \times \mathcal{X} \times \mathcal{A}$ is the transition function, where $P(x'|x, a)$ is the probability that the next state of the Markovian environment will be $x'$, given that action $a$ is selected in state $x$. We will assume that $M$ satisfies the following assumptions:

- The state space $\mathcal{X}$ can be decomposed into non-intersecting layers, i.e. $\mathcal{X} = \bigcup_{k=0}^{L} \mathcal{X}_k$ where $\mathcal{X}_l \cap \mathcal{X}_k = \emptyset$ for $l \neq k$.
- $\mathcal{X}_0$ and $\mathcal{X}_L$ are singletons, i.e. $\mathcal{X}_0 = \{x_0\}$ and $\mathcal{X}_L = \{x_L\}$.

- Transitions are possible only between consecutive layers. Formally, if $P(x'|x,a) > 0$, then $x' \in \mathcal{X}_{k+1}$ and $x \in \mathcal{X}_k$ for some $0 \le k \le L-1$.

The interaction between the learner and the environment is described on Figure 1. The interaction of an agent and the Markovian environment proceeds in episodes, where in each episode the agent starts in state $x_0$ and moves forward across the consecutive layers until it reaches state $x_L$.[1] We assume that the environment selects a sequence of loss functions $\{\ell_t\}_{t=1}^T$ and the losses only change between episodes. Furthermore, we assume that the learner only observes the losses that it suffers in each individual state-action pair that it visits, in other words, we consider *bandit feedback*.[2]

---

**Parameters**: Markovian environment $M = \{\mathcal{X}, \mathcal{A}, P\}$;
**For all episodes** $t = 1, 2, \ldots, T$, **repeat**

    1. The environment chooses the loss function $\ell_t : \mathcal{X} \times \mathcal{A} \to [0, 1]$.

    2. The learner starts in state $\mathbf{x}_0(t) = x_0$.

    3. **For all time steps** $l = 0, 1, 2, \ldots, L-1$, **repeat**

        (a) The learner observes $\mathbf{x}_l(t) \in \mathcal{X}_l$.

        (b) Based on its previous observations (and randomness), the learner selects $\mathbf{a}_l(t)$.

        (c) The learner suffers and observes loss $\ell_t(\mathbf{x}_l(t), \mathbf{a}_l(t))$.

        (d) The environment draws the new state $\mathbf{x}_{l+1}(t) \sim P(\cdot|\mathbf{x}_l(t), \mathbf{a}_l(t))$.

---

Figure 1: The protocol of online learning in episodic MDPs.

For defining our performance measure, we need to specify a set of reference controllers that is made available to the learner. To this end, we define the concept of (stochastic stationary) *policies*: A policy is defined as a mapping $\pi : \mathcal{A} \times \mathcal{X} \to [0, 1]$, where $\pi(a|x)$ gives the probability of selecting action $a$ in state $x$. The expected total loss of a policy $\pi$ is defined as

$$L_T(\pi) = \mathbb{E}\left[ \sum_{t=1}^T \sum_{k=0}^{L-1} \ell_t(\mathbf{x}_k', \mathbf{a}_k') \middle| P, \pi \right],$$

where the notation $\mathbb{E}[\cdot \,|\, P, \pi]$ is used to emphasize that the random variables $\mathbf{x}_k'$ and $\mathbf{a}_k'$ are generated by executing $\pi$ in the MDP specified by the transition function $P$. Denote the total expected loss suffered by the learner as $\widehat{L}_T = \sum_{t=1}^T \sum_{k=0}^{L-1} \mathbb{E}[\ell_t(\mathbf{x}_k(t), \mathbf{a}_k(t))|\, P]$, where the expectation is taken over the internal randomization of the learner and the random transitions of the Markovian environment. Using these notations, we define the learner's goal as minimizing the (total expected) *regret* defined as

$$\widehat{\mathfrak{R}}_T = \widehat{L}_T - \min_\pi L_T(\pi),$$

where the minimum is taken over the complete set of stochastic stationary policies.[3]

It is beneficial to introduce the concept of *occupancy measures* on the state-action space $\mathcal{X} \times \mathcal{A}$: the occupancy measure $q^\pi$ of policy $\pi$ is defined as the collection of distributions generated by executing policy $\pi$ on the episodic MDP described by $P$:

$$q^\pi(x,a) = \mathbb{P}\left[ \mathbf{x}_{k(x)}' = x, \mathbf{a}_{k(x)}' = a \middle| P, \pi \right],$$

where $k(x)$ denotes the index of the layer that $x$ belongs to. It is easy to see that the occupancy measure of any policy $\pi$ satisfies

$$\sum_a q^\pi(x,a) = \sum_{x' \in \mathcal{X}_{k(x)-1}} \sum_{a'} P(x|x',a') q^\pi(x',a'), \tag{1}$$

for all $x \in \mathcal{X} \setminus \{x_0, x_l\}$, with $q^\pi(x_0, a) = \pi(a|x_0)$ for all $a \in \mathcal{A}$. The set of all occupancy measures satisfying the above equality in the MDP $M$ will be denoted as $\Delta(M)$. The policy $\pi$ is said to generate the occupancy measure $q \in \Delta(M)$ if

$$\pi(a|x) = \frac{q(x, a)}{\sum_b q(x, b)}$$

holds for all $(x, a) \in \mathcal{X} \times \mathcal{A}$. It is clear that there exists a unique generating policy for all measures in $\Delta(M)$ and vice versa. The policy generating $q$ will be denoted as $\pi^q$. In what follows, we will redefine the task of the learner from having to select individual actions $\mathbf{a}_k(t)$ to having to select occupancy measures $\mathbf{q}_t \in \Delta(M)$ in each episode $t$. To see why this notion simplifies the treatment of the problem, observe that

$$\mathbb{E}\left[\left.\sum_{k=0}^{L-1} \ell_t(\mathbf{x}'_k, \mathbf{a}'_k)\right| P, \pi^q\right] = \sum_{k=0}^{L-1} \sum_{x \in \mathcal{X}_k} \sum_a q(x, a)\ell_t(x, a)$$
$$= \sum_{x,a} q(x, a)\ell_t(x, a) \stackrel{\text{def}}{=} \langle q, \ell_t \rangle, \tag{2}$$

where we defined the inner product $\langle \cdot, \cdot \rangle$ on $\mathcal{X} \times \mathcal{A}$ in the last line. Using this notation, we can reformulate our original problem as an instance of online linear optimization with decision space $\Delta(M)$. Assuming that the learner selects occupancy measure $\mathbf{q}_t$ in episode $t$, the regret can be rewritten as

$$\widehat{\mathfrak{R}}_T = \max_{q \in \Delta(M)} \mathbb{E}\left[\sum_{t=1}^T \langle \mathbf{q}_t - q, \ell_t \rangle\right].$$

## 3 The algorithm: O-REPS

Using the formalism introduced in the previous section, we now describe our algorithm called Online Relative Entropy Policy Search (O-REPS). O-REPS is an instance of online linear optimization methods usually referred to as Follow-the-Regularized-Leader (FTRL), Online Stochastic Mirror Descent (OSMD) or the Proximal Point Algorithm (PPA)—see, e.g., [1], [19], [3] and [2] for a discussion of these methods and their relations. To allow comparisons with the original derivation of REPS by Peters et al. [17], we formalize our algorithm as an instance of PPA. Before describing the algorithm, some more definitions are in order. First, define $D(q\|q')$ as the unnormalized Kullback–Leibler divergence between two occupancy measures $q$ and $q'$:

$$D(q\|q') = \sum_{x,a} q(x, a) \log \frac{q(x, a)}{q'(x, a)} - \sum_{x,a} (q(x, a) - q'(x, a)).$$

Furthermore, let $R(q)$ define the unnormalized negative entropy of the occupancy measure $q$:

$$R(q) = \sum_{x,a} q(x, a) \log q(x, a) - \sum_{x,a} q(x, a).$$

We are now ready to define O-REPS formally. In the first episode, O-REPS chooses the uniform policy with $\boldsymbol{\pi}_1(a|x) = 1/|\mathcal{A}|$ for all $x$ and $a$, and we let $\mathbf{q}_1 = q^{\boldsymbol{\pi}_1}$.[4] Then, the algorithm proceeds recursively: After observing

$$\mathbf{u}_t = (\mathbf{x}_0(t), \mathbf{a}_0(t), \ell_t(\mathbf{x}_0(t), \mathbf{a}_0(t)), \dots, \mathbf{x}_{L-1}(t), \mathbf{a}_{L-1}(t), \ell_t(\mathbf{x}_{L-1}(t), \mathbf{a}_{L-1}(t)), \mathbf{x}_L(t))$$

in episode $t$, we define the loss estimates $\hat{\boldsymbol{\ell}}_t$ as

$$\hat{\boldsymbol{\ell}}_t = \frac{\ell_t(x, a)}{\mathbf{q}_t(x, a)} \mathbb{I}\{(x, a) \in \mathbf{u}_t\},$$

where we used the notation $(x, a) \in \mathbf{u}_t$ to indicate that the state-action pair $(x, a)$ was observed during episode $t$. After episode $t$, O-REPS selects the occupancy measure that solves the optimization problem

$$\mathbf{q}_{t+1} = \underset{q \in \Delta(M)}{\arg\min} \left\{\eta \left\langle q, \hat{\boldsymbol{\ell}}_t \right\rangle + D(q\|\mathbf{q}_t)\right\}. \tag{3}$$

In episode $t$, our algorithm follows the policy $\boldsymbol{\pi}_t = \pi^{\mathbf{q}_t}$. Defining $\mathbf{U}_t = (\mathbf{u}_1, \mathbf{u}_2, \ldots, \mathbf{u}_t)$, we clearly have that $\mathbf{q}_t(x, a) = \mathbb{P}\left[(x, a) \in \mathbf{u}_t \mid \mathbf{U}_{t-1}\right]$, so $\hat{\ell}_t(x, a)$ is an unbiased estimate of $\ell_t(x, a)$ for all $(x, a)$ such that $\mathbf{q}_t(x, a) > 0$:

$$\mathbb{E}\left[\hat{\ell}_t(x, a) \,\middle|\, \mathbf{U}_{t-1}\right] = \frac{\ell_t(x, a)}{\mathbf{q}_t(x, a)} \mathbb{P}\left[(x, a) \in \mathbf{u}_t \mid \mathbf{U}_{t-1}\right] = \ell_t(x, a). \tag{4}$$

We now proceed to explain how the policy update step (3) can be implemented efficiently. It is known (see, e.g., Bartók et al. [3, Lemma 8.6]) that performing this optimization can be reformulated as first solving the unconstrained optimization problem

$$\tilde{\mathbf{q}}_{t+1} = \arg\min_q \left\{\eta \left\langle q, \hat{\boldsymbol{\ell}}_t \right\rangle + D(q\|\mathbf{q}_t)\right\}$$

and then projecting the result to $\Delta(M)$ as

$$\mathbf{q}_{t+1} = \arg\min_{q \in \Delta(M)} D\left(q\|\tilde{\mathbf{q}}_{t+1}\right).$$

The first step can be simply carried out by setting $\tilde{\mathbf{q}}_{t+1}(x, a) = \mathbf{q}_t(x, a)e^{-\eta\hat{\ell}_t(x,a)}$. The projection step, however, requires more care. To describe the projection procedure, we need to introduce some more notation. For any function $v : \mathcal{X} \to R$ and loss function $\ell : \mathcal{X} \times \mathcal{A} \to [0, 1]$ we define a function

$$\delta(x, a|v, \ell) = -\eta\ell(x, a) - \sum_{x' \in \mathcal{X}} v(x')P(x'|x, a) + v(x). \tag{5}$$

As noted by Peters et al. [17], the above function can be regarded as the Bellman error corresponding to the value function $v$. The next proposition provides a succinct formalization of the optimization problem (3).

**Proposition 1.** *Let $t > 1$ and define the function*

$$\mathbf{Z}_t(v, k) = \sum_{x \in \mathcal{X}_k, a \in \mathcal{A}} \mathbf{q}_t(x, a)e^{\delta(x, a|v, \hat{\boldsymbol{\ell}}_t)}.$$

*The update step* (3) *can be performed as*

$$\mathbf{q}_{t+1}(x, a) = \frac{\mathbf{q}_t(x, a)e^{\delta(x, a|\hat{\mathbf{v}}_t, \hat{\boldsymbol{\ell}}_t)}}{\mathbf{Z}_t(\hat{\mathbf{v}}_t, k(x))},$$

*where*

$$\hat{\mathbf{v}}_t = \arg\min_v \sum_{k=0}^{L} \ln \mathbf{Z}_t(v, k). \tag{6}$$

Minimizing the expression on the right-hand side of Equation (6) is an unconstrained convex optimization problem (see Boyd and Vandenberghe [4] and the comments of Peters et al. [17]) and can be solved efficiently. It is important to note that since $\mathbf{q}_1(x, a) > 0$ holds for all $(x, a)$ pairs, $\mathbf{q}_t(x, a)$ is also positive for all $t > 0$ by the multiplicative update rule, so Equation 4 holds for all state-action pairs $(x, a)$ in all time steps.

The proof follows the steps of Peters et al. [17], however, their original formalization of REPS is slightly different, which results in small changes in the analysis as well. For further comments regarding the differences between O-REPS and REPS, see Section 5.

*Proof of Proposition 1.* We start with formulating the projection step as a constrained optimization problem:

$$\min_q D\left(q\|\tilde{\mathbf{q}}_{t+1}\right)$$

subject to $\quad \sum_a q(x, a) = \sum_{x', a'} P(x|x', a')q(x', a') \qquad$ for all $x \in \mathcal{X} \setminus \{x_0, x_l\}$,

$$\sum_{x \in \mathcal{X}_k} \sum_a q(x, a) = 1 \qquad\qquad\qquad \text{for all } k = 0, 1, \ldots, L-1.$$

To solve the problem, consider the Lagrangian:

$$\mathcal{L}_t(q) = D\left(q\|\tilde{\mathbf{q}}_{t+1}\right) + \sum_{k=0}^{L-1} \lambda_k \left(\sum_{x\in\mathcal{X}_k, a\in\mathcal{A}} q(x,a) - 1\right)$$

$$+ \sum_{k=1}^{L-1} \sum_{x\in\mathcal{X}_k} v(x) \left(\sum_{x'\in\mathcal{X}_{k-1}} \sum_{a'} q(x',a')P(x|x',a') - \sum_a q(x,a)\right)$$

$$= D\left(q\|\tilde{\mathbf{q}}_{t+1}\right) + \sum_a q(x_0,a)\left(\lambda_0 + \sum_{x'} v(x')P(x'|x_0,a)\right) - \sum_{k=0}^{L-1} \lambda_k$$

$$+ \sum_{x\neq x_0} \sum_a q(x,a)\left(\lambda_{k(x)} + \sum_{x'} v(x')P(x'|x,a) - v(x)\right),$$

where $\{\lambda_k\}_{k=0}^{L-1}$ and $\{v(x)\}_{x\in\mathcal{X}\setminus\{x_0,x_l\}}$ are Lagrange multipliers. In what follows, we set $v(x_0) = v(x_L) = 0$ for convenience. Differentiating the Lagrangian with respect to any $q(x,a)$, we get

$$\frac{\partial\mathcal{L}_t(q)}{\partial q(x,a)} = \ln q(x,a) - \ln\tilde{\mathbf{q}}_{t+1}(x,a) + \lambda_{k(x)} + \sum_{x'} v(x')P(x'|x,a) - v(x).$$

Hence, setting the gradient to zero, we obtain the formula for $\mathbf{q}_{t+1}(x,a)$:

$$\mathbf{q}_{t+1}(x,a) = \tilde{\mathbf{q}}_{t+1}(x,a)e^{-\lambda_{k(x)} - \sum_{x'} v(x')P(x'|x,a) + v(x)}.$$

Substituting the formula for $\tilde{\mathbf{q}}_{t+1}(x,a)$, we get

$$\mathbf{q}_{t+1}(x,a) = \mathbf{q}_t(x,a)e^{-\lambda_{k(x)} + \delta(x,a|v,\hat{\boldsymbol{\ell}}_t)}.$$

Using the second constraint, we have for every $k = 0, 1, \ldots, L-1$ that

$$\sum_{x\in\mathcal{X}_k} \sum_a \mathbf{q}_t(x,a)e^{-\lambda_k + \delta(x,a|v,\hat{\boldsymbol{\ell}}_t)} = 1,$$

yielding $e^{-\lambda_k} = 1/\mathbf{Z}_t(v,k)$, which leaves us with computing the value of $v$ at the optimum. This can be done by solving the dual problem of maximizing

$$\sum_{x,a} \tilde{\mathbf{q}}_{t+1}(x,a) - L - \sum_{k=0}^{L-1} \lambda_k$$

over $\{\lambda_k\}_{k=0}^{L-1}$. If we drop the constants and express each $\lambda_k$ in terms of $\mathbf{Z}_t(v,k)$, then the problem is equivalent to maximizing $-\sum_{k=0}^{L-1} \ln\mathbf{Z}_t(v,k)$, that is, solving the optimization problem (6).  □

## 4 Analysis

The next theorem states our main result concerning the regret of O-REPS under bandit feedback. The proof of the theorem is based on rather common ideas used in the analysis of FTRL/OSMD/PPA-style algorithms (see, e.g., [24], Chapter 11 of [5], [1], [12], [2]). After proving the theorem, we also present the regret bound for O-REPS when used in a full information setting where the learner gets to observe $\ell_t$ after each episode $t$.

**Theorem 1.** *Assuming bandit feedback, the total expected regret of O-REPS satisfies*

$$\widehat{\mathfrak{R}}_T \leq \eta|\mathcal{X}||\mathcal{A}|T + \frac{L\log\frac{|\mathcal{X}||\mathcal{A}|}{L}}{\eta}.$$

*In particular, setting $\eta = \sqrt{L\frac{\log\frac{|\mathcal{X}||\mathcal{A}|}{L}}{T|\mathcal{X}||\mathcal{A}|}}$ yields*

$$\widehat{\mathfrak{R}}_T \leq 2\sqrt{L|\mathcal{X}||\mathcal{A}|T\log\frac{|\mathcal{X}||\mathcal{A}|}{L}}.$$

*Proof.* By standard arguments (see, e.g., [19, Lemma 12], [3, Lemma 9.2] or [5, Theorem 11.1]), we have

$$\sum_{t=1}^{T} \left\langle \mathbf{q}_t - q, \hat{\boldsymbol{\ell}}_t \right\rangle \leq \sum_{t=1}^{T} \left\langle \mathbf{q}_t - \tilde{\mathbf{q}}_{t+1}, \hat{\boldsymbol{\ell}}_t \right\rangle + \frac{D\left(q\|\mathbf{q}_1\right)}{\eta}. \tag{7}$$

Using the exact form of $\tilde{\mathbf{q}}_{t+1}$ and the fact that $e^x \geq 1 + x$, we get that

$$\tilde{\mathbf{q}}_{t+1}(x, a) \geq \mathbf{q}_t(x, a) - \eta \mathbf{q}_t(x, a) \hat{\boldsymbol{\ell}}_t(x, a)$$

and thus

$$\sum_{t=1}^{T} \left\langle \mathbf{q}_t - \tilde{\mathbf{q}}_{t+1}, \hat{\boldsymbol{\ell}}_t \right\rangle \leq \eta \sum_{t=1}^{T} \sum_{x,a} \mathbf{q}_t(x, a) \hat{\boldsymbol{\ell}}_t^2(x, a)$$

$$\leq \eta \sum_{t=1}^{T} \sum_{x,a} \mathbf{q}_t(x, a) \frac{\ell_t(x, a)}{\mathbf{q}_t(x, a)} \hat{\boldsymbol{\ell}}_t(x, a) \leq \eta \sum_{t=1}^{T} \sum_{x,a} \hat{\boldsymbol{\ell}}_t(x, a).$$

Combining this with (7), we get

$$\sum_{t=1}^{T} \left\langle \mathbf{q}_t - q, \hat{\boldsymbol{\ell}}_t \right\rangle \leq \eta \sum_{t=1}^{T} \sum_{x,a} \hat{\boldsymbol{\ell}}_t(x, a) + \frac{D\left(q\|\mathbf{q}_1\right)}{\eta}. \tag{8}$$

Next, we take an expectation on both sides. By Equation (4), we have

$$\mathbb{E}\left[ \sum_{t=1}^{T} \sum_{x,a} \hat{\boldsymbol{\ell}}_t(x, a) \right] \leq |\mathcal{X}||\mathcal{A}|T.$$

It also follows from Equation (4) that $\mathbb{E}\left[ \left\langle q, \hat{\boldsymbol{\ell}}_t \right\rangle \right] = \langle q, \ell_t \rangle$ and $\mathbb{E}\left[ \left\langle \mathbf{q}_t, \hat{\boldsymbol{\ell}}_t \right\rangle \right] = \mathbb{E}\left[ \langle \mathbf{q}_t, \ell_t \rangle \right]$. Finally, notice that

$$D\left(q\|\mathbf{q}_1\right) \leq R(q) - R(\mathbf{q}_1) \leq \sum_{k=0}^{L-1} \sum_{x \in \mathcal{X}_k} \sum_a \mathbf{q}_1(x, a) \log \frac{1}{\mathbf{q}_1(x, a)}$$

$$(\text{since } R(q) \leq 0)$$

$$\leq \sum_{k=0}^{L-1} \log |\mathcal{X}_k||\mathcal{A}| \leq L \log \frac{|\mathcal{X}||\mathcal{A}|}{L},$$

where we used the trivial upper bound on the entropy of distributions and Jensen's inequality in the last step. Plugging the above upper bound into Equation (8), we obtain the statement of the theorem. $\qquad \square$

**Theorem 2.** *Assuming full feedback, the total expected regret of O-REPS satisfies*

$$\widehat{\mathfrak{R}}_T \leq \eta LT + \frac{L \log \frac{|\mathcal{X}||\mathcal{A}|}{L}}{\eta}.$$

*In particular, setting $\eta = \sqrt{\frac{\log \frac{|\mathcal{X}||\mathcal{A}|}{L}}{T}}$ yields*

$$\widehat{\mathfrak{R}}_T \leq 2L \sqrt{T \log \frac{|\mathcal{X}||\mathcal{A}|}{L}}.$$

The proof of the statement follows directly from the proof of Theorem 1, with the only difference that we set $\hat{\boldsymbol{\ell}}_t = \ell_t$ and we can use the tighter upper bound

$$\sum_{t=1}^{T} \langle \mathbf{q}_t - \tilde{\mathbf{q}}_{t+1}, \ell_t \rangle \leq \eta \sum_{t=1}^{T} \sum_{x,a} \mathbf{q}_t(x, a) \ell_t^2(x, a)$$

$$\leq \eta \sum_{t=1}^{T} \sum_{x,a} \mathbf{q}_t(x, a) = \eta LT,$$

where we used that $\sum_{x \in \mathcal{X}_k} \sum_a \mathbf{q}_t(x, a) = 1$ for all layers $k$.

# 5 Conclusions and future work

**Comparison with previous results**  We first compare our regret bounds with previous results from the literature. First, our guarantees for the full information case trade off a factor of $L$ present in the bounds of Neu et al. [14] to a (usually much smaller) factor of $\sqrt{\log|\mathcal{X}|}$. More importantly, our bounds trade off a factor of $L^{3/2}/\alpha$ in the bandit case to a factor of $\sqrt{|\mathcal{X}|}$. This improvement is particularly remarkable considering that we do not need to assume that $\alpha > 0$, that is, we drop the rather unnatural assumption that every stationary policy has to visit every state with positive probability. In particular, dropping this assumption enables our algorithm to work in deterministic loop-free MDPs, that is, to solve the online shortest path problem (see, e.g., [10]). In the shortest path setting, O-REPS provides an alternative implementation to the Component Hedge algorithm analyzed by Koolen et al. [12], who prove identical bounds in the full information case. As shown by Audibert et al. [2], Component Hedge achieves the analog of our bounds in the bandit case as well.

O-REPS also bears close resemblance to the algorithms of Even-Dar et al. [9] and Neu et al. [16] who also use policy updates of the form $\boldsymbol{\pi}_{t+1}(a|x) \propto \boldsymbol{\pi}_t(a|x) \exp(-\eta \ell_t(x,a) - \sum_{x'} P(x'|x,a) v_t(x'))$. The most important difference between their algorithm and O-REPS is that their value functions $v_t$ are computed as the solution of the Bellman-equations instead of the solution of the optimization problem (6). By a simple combination of our analysis and that of Even-Dar et al. [9], it is possible to show that O-REPS attains a regret of $\widetilde{O}(\sqrt{\tau T})$ in the unichain setting with full information feedback, improving their bound by a factor of $\tau^{3/2}$ under the same assumptions. It is an interesting open problem to find out if using the O-REPS value functions is a strictly better idea than solving the Bellman equations in general. Another important direction of future work is to extend our results to the case of unichain MDPs with bandit feedback and the setting where the transition probabilities of the underlying MDP is unknown (see Neu et al. [15]).

**Lower bounds**  Following the proof of Theorem 10 in Audibert et al. [2], it is straightforward to construct an MDP consisting of $|\mathcal{X}|/L$ chains of $L$ consecutive bandit problems each with $|\mathcal{A}|$ actions such that no algorithm can achieve smaller regret than $0.03L\sqrt{T \log(|\mathcal{X}||\mathcal{A}|)}$ in the full information case and $0.04\sqrt{L|\mathcal{X}||\mathcal{A}|T}$ in the bandit case. These results suggest that our bounds cannot be significantly improved in general, however, finding an appropriate *problem-dependent* lower bound remains an interesting open problem in the much broader field of online linear optimization.

**REPS vs. O-REPS**  As noted several times above, our algorithm is directly inspired by the work of Peters et al. [17]. However, there is a slight difference between the original version of REPS and O-REPS, namely, Peters et al. aim to solve the optimization problem $\mathbf{q}_{t+1} = \arg\min_{q \in \Delta(M)} \langle q, \hat{\boldsymbol{\ell}}_t \rangle$ subject to the constraint $D(q \| \mathbf{q}_t) \leq \varepsilon$ for some $\varepsilon > 0$. This is to be contrasted with the following property of the occupancy measures generated by O-REPS (proved in the supplementary material):

**Lemma 1.** *For any $t > 0$, $D(\mathbf{q}_t \| \mathbf{q}_{t+1}) \leq \frac{\eta^2}{2} \langle \mathbf{q}_t, \hat{\boldsymbol{\ell}}_t^2 \rangle$.*

In particular, if the losses are estimated by bounded sample averages as done by Peters et al. [17], this gives $D(\mathbf{q}_t \| \mathbf{q}_{t+1}) \leq \eta^2/2$. While this is not the exact same property as desired by REPS, both inequalities imply that the occupancy measures stay close to each other in the 1-norm sense by Pinsker's inequality. Thus we conjecture that our formulation of O-REPS has similar properties to the one studied by Peters et al. [17], while it might be somewhat simpler to implement.

**Acknowledgments**

Alexander Zimin is an OMV scholar. Gergely Neu's work was carried out during the tenure of an ERCIM "Alain Bensoussan" Fellowship Programme. The research leading to these results has received funding from INRIA, the European Union Seventh Framework Programme (FP7/2007-2013) under grant agreements 246016 and 231495 (project CompLACS), the Ministry of Higher Education and Research, Nord-Pas-de-Calais Regional Council and FEDER through the "Contrat de Projets Etat Region (CPER) 2007-2013".

## Footnotes

[1] Such MDPs naturally arise in episodic decision tasks where some notion of time is present in the state description.

[2] In the literature of online combinatorial optimization, this feedback scheme is often called *semi-bandit feedback*, see Audibert et al. [2].

[3] The existence of this minimum is a standard result of MDP theory, see Puterman [18].

[4]Note that $q^\pi$ can be simply computed by using (1) recursively.

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
