[Supplementary Material]

*Proof of Lemma 1.* We start with expanding $D\left(\mathbf{q}_t \| \mathbf{q}_{t+1}\right)$:

$$
\begin{aligned}
D\left(\mathbf{q}_t \| \mathbf{q}_{t+1}\right) &= \sum_{x,a} \mathbf{q}_t(x,a) \ln \frac{\mathbf{q}_t(x,a)}{\mathbf{q}_{t+1}(x,a)} \\
&= \sum_{x,a} \mathbf{q}_t(x,a) \ln \frac{\mathbf{Z}_t(\hat{\mathbf{v}}_t, k)}{e^{\delta(x,a|\hat{\mathbf{v}}_t, \hat{\boldsymbol{\ell}}_t)}} \\
&= \sum_{k=0}^{L-1} \ln \mathbf{Z}_t(\hat{\mathbf{v}}_t, k) - \sum_{x,a} \mathbf{q}_t(x,a) \delta(x,a|\hat{\mathbf{v}}_t, \hat{\boldsymbol{\ell}}_t). \quad (1)
\end{aligned}
$$

By Proposition 1, we have that

$$
\begin{aligned}
\sum_{k=0}^{L-1} \ln \mathbf{Z}_t(\hat{\mathbf{v}}_t, k) &\leq \sum_{k=0}^{L-1} \ln \mathbf{Z}_t(0, k) = \sum_{k=0}^{L-1} \ln \sum_{x \in \mathcal{X}_k} \sum_a \mathbf{q}_t(x,a) e^{-\eta \hat{\ell}_t(x,a)} \\
&\leq \sum_{k=0}^{L-1} \ln \left( 1 - \eta \sum_{x \in \mathcal{X}_k} \sum_a \mathbf{q}_t(x,a) \hat{\ell}_t(x,a) + \sum_{x \in \mathcal{X}_k} \sum_a \mathbf{q}_t(x,a) \frac{(-\eta \hat{\ell}_t(x,a))^2}{2} \right) \\
&\leq -\eta \sum_{x,a} \mathbf{q}_t(x,a) \hat{\ell}_t(x,a) + \sum_{x,a} \mathbf{q}_t(x,a) \frac{(-\eta \hat{\ell}_t(x,a))^2}{2}
\end{aligned}
$$

where we used the facts that $e^s \leq 1 + s + \frac{s^2}{2}$ holds for all $s \leq 0$ and $\log(1 + s) \leq s$ holds for all $s \in \mathbb{R}$.

The second term in (1) is rewritten as follows:

$$
\begin{aligned}
\sum_{x,a} q_t(x,a) \delta(x,a|\hat{\mathbf{v}}_t, \hat{\boldsymbol{\ell}}_t) = &-\eta \sum_{x,a} \mathbf{q}_t(x,a) \hat{\ell}_t(x,a) \\
&+ \sum_{x,a} \mathbf{q}_t(x,a) \hat{\mathbf{v}}_t(x) - \sum_{x,a} \mathbf{q}_t(x,a) \sum_{x' \in \mathcal{X}_{k(x)+1}} \hat{\mathbf{v}}_t(x') P(x'|x,a).
\end{aligned}
$$

By the property of the occupancy measure, we have for all $k = 0, 1, \ldots, L-1$ that

$$
\begin{aligned}
\sum_{x \in \mathcal{X}_k} \sum_a \mathbf{q}_t(x,a) \sum_{x' \in \mathcal{X}_{k+1}} \hat{\mathbf{v}}_t(x') P(x'|x,a) &= \sum_{x' \in \mathcal{X}_{k+1}} \hat{\mathbf{v}}_t(x') \sum_{x \in \mathcal{X}_k} \sum_a \mathbf{q}_t(x,a) P(x'|x,a) \\
&= \sum_{x' \in \mathcal{X}_{k+1}} \hat{\mathbf{v}}_t(x') \sum_a \mathbf{q}_t(x', a) \\
&= \sum_{x \in \mathcal{X}_{k+1}} \sum_a \mathbf{q}_t(x,a) \hat{\mathbf{v}}_t(x).
\end{aligned}
$$

Since $\hat{\mathbf{v}}_t(x_0) = \hat{\mathbf{v}}_t(x_L) = 0$, we have

$$
\sum_{x,a} \mathbf{q}_t(x,a) \delta(x,a|\hat{\mathbf{v}}_t, \hat{\boldsymbol{\ell}}_t) = -\eta \sum_{x,a} \mathbf{q}_t(x,a) \hat{\ell}_t(x,a).
$$

Combining the obtained expressions for the terms on the right-hand side of (1), we obtain the statement of the lemma as

$$
D\left(\mathbf{q}_t \| \mathbf{q}_{t+1}\right) \leq \frac{\eta^2}{2} \sum_{x,a} \mathbf{q}_t(x,a) \hat{\ell}_t^2(x,a).
$$

$\square$