[Reviews · NeurIPS 2013]

Submitted by Assigned_Reviewer_4

Overall, the paper is well-written. I suggest adding a bit more high-level
explanation/intuition to separate your approach from the earlier technique
of decomposing into several learning problems (one for each state of
Even-Dar et al.). Also, in Section 5.1 you mention that the techniques in
your paper could be applied to get better regret bounds than those of
Even-Dar et al. Is this true under the conditions they consider in their
paper? It would be good to elaborate on this point a bit more.

A minor comment:
---------------------
1. In Fig. 1, Line 3(b) : Instead of saying, "the learner (randomly)
selects", I'd say "Based on previous observations (and randomness), the
learner selects" -- to emphasize that the learner is doing something
smart.


Summary: The paper considers an episodic version of online MDP problem. Here, at
each episode there is a L-layer MDP with a fixed start and finish state.
Within each episode there are L time-steps, where an action is taken,
state changes and the loss/reward is observed. The version considered is
the so-called semi-bandit setting, where the loss is observed at each
time-step within the episode, but only corresponding to the actual action
chosen. An algorithm with almost optimal regret bounds is presented, which
can also be implemented in polynomial time.

Submitted by Assigned_Reviewer_5

The authors study what they refer to as an episodic loop-free MDP, with adversarial losses selected between each episode. This is an interesting class of MDPs for applications where time spent in an episode is part of the state information available to the learner. In other words, the MDP is layered, with actions taken in layer k leading to states in layer k+1. At the final layer L, a new set of losses is adversarially selected, and the learner begins anew at the 0th layer.

The setting considered by the authors is essentially that of Neu et al (http://colt2010.haifa.il.ibm.com/papers/125neu.pdf), except they remove the assumption that policies must visit a state with some positive probability alpha. In the bandit setting, the resulting regret bounds shave off the factor of 1/alpha present in Neu et al., which is significant. They also get a sqrt{LT} dependence, instead of the L^2\sqrt{T} dependence in prior work.

The authors cast the problem as online linear optimization by considering an algorithm that selects a fixed distribution over actions, for each state, at the start of an episode. Their algorithm is inspired by Relative Entropy Policy Search algorithm of Peters et al, and works by selecting the distribution that minimizes the previous episode's loss estimates and a KL regularization term. Full proofs are provided and appear to be correct. They also comment that their analysis, in either the full information or bandit setting, cannot be significantly improved in general, as they attain the lower bound for a construction by Audibert et al.

The results in the paper seem to be correct. It is clear, well presented, and significant.

If the authors are not already planning to do so, I would recommend citing Dekel et al. from the most recent ICML who handle generic adversarial MDPs, but with deterministic transitions.
Summary: This is a well-written paper that provides a significant improvement over existing algorithms for a specific class of MDPs with adversarial losses.

Submitted by Assigned_Reviewer_6

Summary
The paper is about online learning in non-stationary episodic loop-free Markov Decision Processes, where the loss function is allowed to change between episodes.
The proposed algorithm O-REPS is an online variant of the Relative Entropy Policy Search algorithm, that searches for better policies by bounding the related information loss, so that small policies updates are performed.
The authors describe the proposed algorithm and produce upper bounds to the total expected regret both for the bandit and for the full feedback cases, concluding that, in many common settings, O-REPS has better guarantees than previous results from literature and requires less assumptions on the MDP and puts no restriction to the class of policies that must be considered.

Quality
The paper is technically sound. The problem is well introduced and the main results are presented with proofs (I have checked them and they seem correct).
In the last section, the authors compare the properties of the proposed approach w.r.t. the ones of state-of-the-art algorithms, highlighting the improvements introduced.
Although no limitation of the proposed approach is discussed, some ideas for future research directions are suggested.

Clarity
The paper is well written and clearly organized. The proofs are quite detailed, so that they can be easily followed in most cases. I suggest only to add some justification to inequalities in line 353.
The only part that I found not very clear is Section 5.3 where the authors tries to compare the proposed approach with the original REPS algorithm.
Overall, although the paper is quite technical, I think that it can be followed and mostly understood also by a non-expert audience.

Originality
The paper builds upon on a state-of-the-art algorithm, but it considers a different setting and produce a theoretical analysis of the algorithm properties that, as far as I know, constitutes an original contribution.
The authors compare the obtained results with the ones obtained by other approaches, giving a quite comprehensive overview of the related literature.
The paper is not groundbreaking, but represents a solid contribution that advances knowledge in a relevant area of machine learning.

Significance
The theoretical results reported in this paper represent a clear improvement over the state of the art.
The impact of such theoretical results on future research can be hardly assessed due to the specific class of MDPs addressed (episodic and loop-free). On the other hand, it provides interesting results that may be useful to produce new studies in more general settings.

*** Comments after the rebuttal ***

Clarifications provided by author rebuttals will further improve paper quality.
Summary: A theoretically sound paper that extends an existing policy search algorithm to the online setting, with the result of obtaining upper bounds to the total regret that are significantly better than the ones previously produced by the literature for episodic loop-free MDPs.
Author Feedback

Author rebuttal: We thank all the reviewers for their comments, and will incorporate their suggestions in our final version. Our replies for each reviewer follow below.

Reviewer 4
----------
Our approach is quite different from the decomposition approach used by Even-Dar et al. since we are not trying to minimize the individual regrets in each state, but rather solve a global optimization problem. Accordingly, there is a slight difference between the resulting algorithms as well: while both algorithms apply exponential updates to the policies, the exponents are not exactly the same. Even-Dar et al. (and Neu et al.) use the action-value functions obtained as the solution of the Bellman equations, while we use the Bellman errors with respect to some value functions that are computed in a different way. It is easy to see that the updates of Even-Dar et al. can be replaced by the Bellman errors with respect to their Bellman-style value functions, so the only difference between the two algorithms lies in how the value functions are computed. We will elaborate on this issue a bit more in our final version.

Also, the result concerning unichain MDPs works precisely under the assumptions of Even-Dar et al. We decided not to include the formal statement of this result since it requires introducing a large number of additional concepts and adds little value to our discussion. Nevertheless, we will point the readers to the paper Even-Dar et al. for the definitions necessary to interpret this result.

Reviewer 5
----------
That is correct, the Dekel et al. paper belongs in our related work section, thank you!

Reviewer 6
----------
We will expand the discussion concerning the relations with the original REPS algorithm.

Also, the equation in line 353 is indeed a bit convoluted, the steps are as follows: 1. using a property of Bregman divergences, 2. omitting the first negative entropy term, and 3. using the fact that q_1 forms a set of distributions defined over each layer, and this term is maximized by picking the uniform distributions over the layers. The inequality is made complete by using Jensen's inequality. These details will be added to our final version.